# Comparative Study of Antimicrobial Activity of Silver, Gold, and Silver/Gold Bimetallic Nanoparticles Synthesized by Green Approach

**DOI:** 10.3390/molecules27227895

**Published:** 2022-11-15

**Authors:** Naila Sher, Dalal Hussien M. Alkhalifah, Mushtaq Ahmed, Nadia Mushtaq, Faridullah Shah, Fozia Fozia, Rahmat Ali Khan, Wael N. Hozzein, Mourad A. M. Aboul-Soud

**Affiliations:** 1Department of Biotechnology, University of Science and Technology, Bannu 28100, Pakistan; 2Department of Biology, College of Science, Princess Nourah bint Abdulrahman University, P.O. Box 84428, Riyadh 11671, Saudi Arabia; 3Department of Botany, University of Science and Technology, Bannu 28100, Pakistan; 4Department of Biochemistry, Rehman Medical Institute, Peshawar 25000, Pakistan; 5Department of Biochemistry, KMU Institute of Medical Sciences, Kohat 26000, Pakistan; 6Botany and Microbiology Department, Faculty of Science, Beni-Suef University, Beni-Suef 62521, Egypt; 7Department of Clinical Laboratory Sciences, College of Applied Medical Sciences, King Saud University, P.O. Box 10219, Riyadh 11433, Saudi Arabia

**Keywords:** *Hippeastrum hybridum*, nanotechnology, NPs synthesis, antimicrobial activities

## Abstract

Nanotechnology is one of the most recent technologies. It is uncertain whether the production of small-size nanoparticles (NPs) can be achieved through a simple, straightforward, and medicinally active phytochemical route. The present study aimed to develop an easy and justifiable method for the synthesis of Ag, Au, and their Ag/Au bimetallic NPs (BNPs) by using *Hippeastrum hybridum* (HH) extract, and then to investigate the effects of Ag, Au, and their Ag/Au BNPs as antimicrobial and phytotoxic agents. Ag, Au, and their Ag/Au BNPs were characterized by UV-visible spectroscopy, FT-IR spectroscopy, XRD, EDX, and SEM analysis. XRD analysis conferring to the face of face-centered cubic crystal structure with an average size of 13.3, 10.72, and 8.34 nm of Ag, Au, and Ag/Au BNPs, respectively. SEM showed that Ag, Au, and Ag/Au BNPs had spherical morphologies, with calculated nano measurements of 40, 30, and 20 nm, respectively. The EDX analysis confirmed the composition of elemental Ag signal of the HH-AgNPs with 22.75%, Au signal of the HH-AuNPs with 48.08%, Ag signal with 12%, and Au signal with 38.26% of the Ag/Au BNPs. The Ag/Au BNPs showed an excellent antimicrobial efficacy against Gram-positive *Staphylococcus aureus*, *Actinomycetes meriye*, *Bacillus cereus*, *Streptococcus pyogenes*, *Methicillin-resistant Staphylococcus aureus*, *Micrococcus luteus*, *Streptococcus pneumonia*, and Gram-negative *Klebsiella pneumonia*, *Escherichia coli*, and *Serratia marcescens* bacterial strains, as well as against three fungal strains (*Aspergillus niger*, *Aspergillus fumigatus*, and *Aspergillus flavus*) compared to HH extract, HH-AgNPs, and HH-AuNPs. However, further investigations are recommended to be able to minimize potential risks of application.

## 1. Introduction

The green synthesis (phytosynthesis) of nanoparticles (NPs) is one of the emergent fields in nanotechnology and nanoscience. Different properties such as electrical, physical, optical, and chemical significantly affect the size and shapes of NPs. Nowadays, NPs, mostly silver (Ag) and gold (Au), are extensively used in biomedicine [1], drug delivery [2], agriculture [3,4], antioxidants [5], as anticancers [6], antibacterials [7], antifungals [8], and ant biofilms agents [9]. For centuries, silver nanoparticles (AgNPs) have been the most widely used antibacterial agents in the healing of wounds, bandages, and medication. [10]. Currently, due to the characteristic antimicrobial potential, AgNPs are also used in other fields, such as medical instrumental devices, cosmetic products, packaging, textiles, food processing, and biosensors [11,12,13]. Apart from AgNPs, gold nanoparticles (AuNPs) are also among the most valuable metals, due to their unique optical, catalytic, and physical properties [14]. The catalytic properties of MNPs can be enhanced to a greater extent by mixing them with some other metals, such as Ag, Ni, Co, and Au, etc. However, bimetallic nanoparticles (BNPs) of Ag and Au metals are of great interest due to their dual influence which can modify their applications. The Ag/Au BNPs showed improved potential as catalytic, optical, electronic, and biomedical applications compared to single metals. BNPs are more important due to the presence of extra degree of freedom. BNPs have shown greater surface area, which increases their adsorption power and means they act as efficient catalysts compared to those of MNPs. BNPs have characteristic mixing patterns and geometrical architecture, which enhances their functionality [15,16].

NPs possess a strong toxicity, which helps to reduce pathogens in a wound, and also possess anti-inflammatory properties which promote nano crystalline bandages to heal chronic ulcers. Recently, different pathogenic microbes of humans have developed resistance to the commercially antimicrobial drugs that are commonly employed in the management of infectious diseases. This undesirable side effect of antibiotics has forced researchers to look for new favorable antimicrobial agents [17]. Currently, metal and their oxides NPs have been used to inhibit the growth of microbes [18,19,20]. The green synthesis AgNPs suggests high antimicrobial activity against both fungal and bacterial pathogens [21]. Similarly, the green-synthesized gold nanoparticles (AuNPs) also demonstrate excellent antimicrobial capabilities [22].

Green synthesis is the method by which the NPs are synthesized using plant extract as a reducing and stabilizing mediator. These plant mediators include Citric acid, saponins, flavonoids, tannins, phenols, Ascorbic acids, dehydrogenases, and extracellular electron shuttles that directly reduce the metal ions into stable NPs. Nowadays, plant-based synthesis of Ag, Au NPs, and Ag/Au BNPs is emergent because of its simplicity, cost-effectiveness, and environmentally friendly synthesis protocols [23,24,25,26].

*Hippeastrum hybridum* (HH) belongs to the Amaryllidaceae family. It is an ornamental bulbous flowering plant, commonly known as Royal Dutch Amaryllis [27]. In ecological and physiological research this plant is commonly used, but the scope of its genomic properties remains limited [28]. HH and other members of the family contain alkaloids, such as montanine, and tazetine [29]. The present research work was conducted on HH extract-induced Ag, Au, and Ag/Au BNPs synthesis, their characterization (UV-visible spectroscopy, FT-IR, XRD, SEM, and EDX), and then their antimicrobial potential. 

## 2. Results

### 2.1. UV-Visible Spectrophotometric Analysis of NPs

In the present research study, we observed the visual reduction of Ag, Au, and Ag/Au BNPs from the addition of AgNO_3_, HAuCl_4_·3H_2_O, and AgNO_3_/HAuCl_4_·3H_2_O (1:1) to HH aqueous solution. The initially uncolored solution changed to brown, ruby red, and reddish-brown colors, indicating Ag, Au, and Ag/Au BNPs formation, respectively. The spectra recorded for the HH plant extract did not exhibit any absorption peak between 200 to 800 nm region, while absorption peaks at 417, 576, and 542 nm, which correspond to Ag, Au, and Ag/Au BNPs, were recorded during the rapid reduction process after 24 h. The difference in the peaks of plant and NPs is due to the reduction of the phytochemicals with the salt Ag and Au ions (Figure 1). 

### 2.2. Factors Effecting Synthesis of NPs

The plant extract, induced HH-AgNPs, HH-AuNPs, and Ag/Au BNPs solutions were exposed to UV-vis spectroscopy during the reduction of ions by HH extract to study the mechanism of Ag, Au, and Ag/Au BNPs formation. The recorded spectra at different pH, such as 4, 5, 6, 7, 8, 9, 10, 11, and 12, are shown in Appendix A. The spectra recorded during the reduction of ions at various pH showed the maximum absorbance with a sharp peak of 417 nm at pH 12, 576 nm at pH 4, and 542 nm at pH 6, which corresponds to the Ag, Au, and Ag/Au BNPs, respectively. The spectra recorded at different concentrations of HH plant, such as 0.5, 1, 1.5, and 2 mL, are shown in Appendix A. The recorded spectra at different HH plant concentrations showed the maximum absorbance with a sharp peak of 417 nm, 576 nm, and 542 nm at 1 mL concentration of plant, which corresponds to the Ag, Au, and Ag/Au BNPs. The UV-vis spectra of Ag, Au, and Ag/Au bimetallic NPs in an aqueous medium at different AgNO_3_/HAuCl_4_·3H_2_O concentrations (0.25, 0.5, 1, and 1.5 mM) were noted at the range of 200 to 800 nm wavelength, which indicated the sharp peaks of 417 nm, 576 nm, and 542 nm at 1 mM salt concentration, which corresponds to the Ag, Au, and Ag/Au BNPs (Appendix A). In the present research work, NPs were studied at different temperature ranges (20 to 100 °C). Broadband appeared at a lower temperature of 20 °C for Ag, Au, and Ag/Au BNPs, while sharp bands of 417 nm, 576 nm, and 542 nm, which correspond to the Ag, Au, and Ag/Au BNPs, appeared at 40 °C; however, with further increase in temperature beyond 40 °C, the band becomes broad and final broadband with lower absorbance appeared at 100 °C (Appendix A). Appendix A indicated the reaction time effect on HH-reduced Ag, Au, and Ag/Au BNPs. A broad peak with lower absorption appeared after 1, 2, and 3 h of the stirring. Due to the continuous reduction of Ag and Au ions by HH extract, the absorption peak increased with time. A final, clear, sharp peak of 417 nm, 576 nm, and 542 nm corresponds to the Ag, Au, and Ag/Au BNPs observed after 24 h of the HH and Ag/Au ions’ reaction. The stability of the HH-reduced Ag, Au, and Ag/Au BNPs was studied at different periods (1 day, 3 months, and 6 months). Sharp peaks with maximum absorbance are reported after 1 day and 3 months of NPs formation; however, after 6 months these peaks became broad and also showed low absorbance for all three types of NPs (Appendix A).

### 2.3. FT-IR Analysis of NPs

The FTIR spectrum analysis of HH extracts Ag, Au, and Ag/Au BNPs (prepared in water) is presented by Figure 2, Appendix A, Figure 3, and Appendix A, respectively. The peak values data and their corresponding functional groups existing in the HH, Ag, Au, and Ag/Au BNPs are presented in Table 1. The characteristic absorption bands were exhibited in the range 3400–3200 cm^−1^ (for O-H stretch), 2935–2915 cm^−1^ (for –CH (CH_2_) vibration), 2865–2845 cm^−1^ (for –CH (CH_2_), 2260–2100 cm^−1^ (for C≡C stretch), 2100–1800 cm^−1^ (for C=O frequency), 1740–1725 cm^−1^ (for C=O stretch), 1650–1600 cm^−1^ (for C=O stretch), 1410–1310 cm^−1^ (for O-H bend), 1340–1250 cm^−1^ (for CN stretch), 995–850 cm^−1^ (for P-O-C stretch), 800–700 cm^−1^ (for C-Cl stretch), 700–600 cm^−1^ (for C-Br stretch), 690–550 cm^−1^ (for C-Br stretch) by HH extract, Ag, Au, and Ag/Au BNPs.

### 2.4. XRD Analysis of NPs

XRD is a technique that is used for determining the size and crystalline nature of the sample. In the present study, the Ag, Au, and Ag/Au BNPs were analyzed by XRD. Figure 4 indicated four Bragg reflections at angles of 37.92°, 43.79°, 64.27°, and 77.18° for AgNPs, Figure 5 showed 37.02°, 44.29°, 64.37°, and 77.58° for AuNPs, and Appendix A showed 38.12°, 44.09°, 64.76°, and 77.48° for Ag/Au BNPs, which are related to the planes (1 1 1), (2 0 0), (2 2 0), and (3 1 1), respectively. These resulting reflections can be indexed conferring to the face of face-centered cubic crystal structures of Ag and Au ions. The “d” (interplanar spacing) and “a” (Miller constants) values were calculated by using the Debye-Sherrer’s Equations (1) and (2), respectively; results are tabulated in Table 2.
(1)$$d_{\mathrm{hkl}}=\frac{\pi }{2{\sin\theta }_{\mathrm{hkl}}}$$
(2)$$å=d_{\mathrm{hkl}}\sqrt{h^{2}+k^{2}+l^{2}}$$

The average crystalline size of HH-induced NPs is calculated by using Debye-Sherrer’s Formula (3);
(3)D=kλ/βcosθ

“D” is the average crystalline size, k is a geometric factor (0.9), λ is the wavelength of the X-ray radiation source, and β is the angular full width at half maximum (FWHM) of the XRD peak at the diffraction angle θ. For the four major peaks, the average crystalline size for each of the samples Ag, Au, and Ag/Au BNPs was found to be 13.3, 10.72, and 8.34 nm, respectively. 

### 2.5. EDX Analysis of NPs

The elemental constituents and relative abundance of the biosynthesized Ag, Au, and Ag/Au BNPs were obtained from EDX analysis, as shown in Figure 6A–C. EDX analysis established the existence of elemental Ag signal of the AgNPs with 22.75%, Au signal of the AuNPs with 48.08%, and Ag signal with 12%/Au signal with 38.26% of the Ag/Au BNPs. The vertical axis displays the number of X-ray counts, while the horizontal axis displays energy in KeV. Identification lines for the major emission energies for Ag and Au are displayed, and these correspond with peaks in the spectrum, thus giving confidence that Ag and Au have been correctly identified in Ag, Au, and Ag/Au BNPs. Thus, the EDX analysis unveils the complete chemical alignment and purity of HH-induced Ag, Au, and Ag/Au BNPs, while the other elements served as coating organic agents bound on the Ag, Au, and Ag/Au BNPs surface. See Table 3 for elemental composition of NPs.

### 2.6. SEM Analysis of Ag, Au, MNPs and Ag-Au BNPs

Figure 7A–C indicated the SEM analysis of Ag, Au, and Ag/Au BNPs synthesized from HH extract: The SEM analysis showed relatively spherical shaped NPs. The morphology of NPs synthesized by the HH extract was confirmed by the SEM analysis. The size was calculated by Nano Measurer 1.2 software by marking 15 particles; size for each of the samples Ag, Au, and Ag/Au BNPs was found to be 40, 30, and 20 nm, respectively. 

### 2.7. Antibacterial Activity

The AgNO_3_ salt, HAuCl_4_·3H_2_O salt, HH extracts, HH-AgNPs, HH-AuNPs, and BNPs showed antibacterial activities at various doses (30, 60, and 90 µg/mL). At the highest 90 µg/mL concentration the HH extracts, HH-AgNPs, HH-AuNPs, and BNPs showed inhibition of 8.5 ± 0.04, 10.5 ± 0.03, 14 ± 0.22, and 15 ± 0.12 mm against *S. aureus*. Against *K. pneumonia* HH extracts, HH-AgNPs, HH-AuNPs, and BNPs showed 2.5 ± 0.11, 5.5 ± 0.1, 9 ± 0.21, and 9.6 ± 0.12 mm inhibition. Against *A. meriye* HH extracts, HH-AgNPs, HH-AuNPs, and BNPs exhibited 3.5 ± 0.012, 9.5 ± 0.1, 6.5 ± 0.05, and 11.5 ± 0.11 mm inhibition. Against *S. pyogen* HH extracts, HH-AgNPs, HH-AuNPs, and BNPs showed 0, 6 ± 23, 9 ± 0.23, and 10 ± 0.08 mm zone inhibition. Against *E. coli* HH extracts, HH-AgNPs, HH-AuNPs, and BNPs showed 4.5 ± 0.011, 6.5 ± 0.17, 7.5 ± 0.12, and 7.8 ± 0.01 mm inhibition. Against *S. marcescens* HH extracts, HH-AgNPs, HH-AuNPs, and BNPs exhibited 2 ± 0.011, 3.5 ± 0.012, 7 ± 0.15, 12.6 ± 0.11 mm inhibition, respectively. Against *B. cereus* HH extracts, HH-AgNPs, HH-AuNPs, and BNPs showed 5.5 ± 0.01, 7 ± 0.03, 2.5 ± 0.11, and 9.5 ± 0.13 mm zone inhibition. Against *MRSA* HH extracts, HH-AgNPs, HH-AuNPs, and BNPs exhibited 6.5 ± 0.011, 6 ± 0.21, 7.5 ± 0.13, and 8.5 ± 0.11 mm inhibition. Against *M. luteus* 7 ± 0.014, 8.5 ± 0.23, 8 ± 0.11, and 10 ± 0.34 mm inhibition were reported for HH extracts, HH-AgNPs, HH-AuNPs, and BNPs, respectively. Against *S. pneumonia* HH extracts, HH-AgNPs, HH-AuNPs, and BNPs showed 4 ± 0.011, 8.5 ± 0.11, 5.5 ± 0.11, and 9 ± 0.11 mm zone inhibition, respectively. AgNO_3_ salt exhibited no zone inhibition against *S. aureus*, *K. pneumonia*, *B. cereus*, and *S. pneumonia*, while 3 ± 0.11, 2.4 ± 0.11, 3.1 ± 0.1, 2 ± 0.13, 0.2 ± 0.11, and 4 ± 0.11 mm inhibition reported against *A. meriye*, *S. pyogen*, *E. coli*, *S. marcescens*, *MRSA*, and *M. luteus,* respectively; similarly, HAuCl_4_·3H_2_O showed no zone inhibition against *S. aureus*, *K. pneumonia*, *A. meriye*, *S. marcescens*, *B. cereus*, *MRSA*, *M. luteus*, and *S. pneumonia*, while 2 ± 0.11 mm inhibition was reported against *S. pyogen* at the highest 90 µg/mL concentration (Appendix A). DMSO served as negative control and levofloxacin (1 g/mL) as a positive control. See Table 4A–F.

### 2.8. Antifungal Activity

From the current study, it has been found that AgNO_3_ salt, HAuCl_4_·3H_2_O salt, HH extracts, HH-AgNPs, HH-AuNPs, and BNPs significantly inhibited the growth of tested fungal strains and proved a valuable antifungal source. At 20 µg/mL, no growth inhibition of *A. niger* and *A. flavus* was reported by AgNO_3_ and HAuCl_4_·3H_2_O salt, while 7 ± 0.12 and 10 ± 0.21 mm inhibition was reported against *A. fumigatus* by AgNO_3_ and HAuCl_4_·3H_2_O salt, respectively. At the highest concentration, 100 µg/mL, AgNO_3_ showed 15 ± 0.11, 31 ± 0.11, and 33 ± 0.14 mm inhibition against *A. niger*, *A. fumigatus*, and *A. flavus,* respectively; similarly, HAuCl_4_·3H_2_O salt showed 11 ± 0.21, 37 ± 04, and 27 ± 0.13 mm inhibition against *A. niger*, *A. fumigatus*, *and A. flavus*, respectively. Consequently, higher growth inhibition of 67 ± 0.13, 75 ± 0.13, 82.5 ± 0.18, and 93 ± 0.18 mm against *A. niger*, 70 ± 0.11, 111 ± 0.15, 94.5 ± 0.21, and 114 ± 0.21 mm against *A. fumigatus*, and 80 ± 0.08, 92.5 ± 0.13, 92 ± 0.04, 112 ± 0.04 mm against *A. flavus* was reported by HH extracts, HH-AgNPs, HH-AuNPs, and BNPs, respectively, at the highest 100 µg/mL concentration (Appendix A). DMSO served as a negative control. No fungal growth was observed in positive control test tubes (terbinafine). See Table 5A–F.

## 3. Materials and Methods

### 3.1. Materials

The *Hippeastrum hybridum* plant was collected in April from Bannu Khyber Pakhtunkhwa (city), and was identified by a taxonomist, Dr. Tahir Iqbal, at the department of Botany, University of Science and Technology, Bannu. Ten clinically isolated bacterial strains, seven Gram-positive *Actinomycetes meriye* ATCC 35568, *Streptococcus pyogenes ATCC* 700294, *Bacillus cereus* ATCC 14579, *Staphylococcus aureus* ATCC 29213, *Methicillin-resistant Staphylococcus aureus* ATCC 43300, *Micrococcus luteus* ATCC 4698, *Streptococcus pneumonia* ATCC 6303, and three Gram-negative *Klebsiella pneumonia* ATCC 700603, *Escherichia coli* ATCC BAA-2471, and *Serratia marcescens* ATCC BAA-3111 were used. Three fungus strains, *Aspergillus fumigatus*, *Aspergillus niger*, and *Aspergillus flavus*, were obtained from the microbiology lab of Biotechnology. Brine shrimps, sea salts, nutrient agar, and Sabourad Dextrose Agar (SDA) were obtained from Sigma. All other reagents used were of analytical grade.

### 3.2. Plant’s Extraction

After identification, the *Hippeastrum hybridum* plant was washed using water, shade dried, and ground to a fine powder. About 250 g of the fine plant powder was mixed with 70% methanol in 1:3, and then retained for extraction at room temperature for 7 days. After that, it was filtered by using Whatman No 1 filter paper, after which the methanol was vaporized at 37 °C to obtain a pure crude methanolic extract of sample, which was then kept in the refrigerator at 4 °C for more studies [30]. 

### 3.3. Synthesis of Ag, Au, and Ag/Au BNPs

For the synthesis of Ag, Au, and Ag-Au BNPs from the HH extract, the standard protocol of [31] was followed with minor modification. Solutions of about 10 mM (0.01 M) of AgNO_3_ and HAuCl_4_·3H_2_O were prepared in 50 mL deionized water. The 10 mM AgNO_3_ and HAuCl_4_·3H_2_O solutions were further diluted 10 times to obtain 1 mM AgNO_3_ and HAuCl_4_·3H_2_O solutions. 0.1 M NaOH, ≥98%, and 0.1 M HCl were used for pH regulation. The HH plant extract was prepared in deionized water (1 gm/100 mL) and agitated quietly for about 1 h on a magnetic stirrer for complete dissolution. The solution was then centrifuged at 6000 rpm for 30 min to obtain the desired product. The supernatant was reserved for further use in NPs synthesis. About 50 mL of the plant supernatant was mixed with 500 mL of: 1 mM AgNO_3_ (pH 12), 1 mM HAuCl_4_·3H_2_O (pH 4), and AgNO_3_/HAuCl_4_·3H_2_O (1:1) solution (pH 6). After 4 h, the reduction of Ag, Au, and Ag/Au BNPs was detected visually using color change. For the complete settlement and stabilization of NPs, the resultant solution was stored for 24 h at room temperature. After that, the NPs were monitored using a UV–Visible spectrophotometer. The colloidal suspension was centrifuged at −4 °C at 10,000 rpm by cold centrifuge for about 10 min; the supernatants were discarded and the pellet (containing NPs) was lyophilized to obtain the powder form. The powder was further characterized and tested for different biological activities.

### 3.4. Factors Affecting Synthesis Rate, Size, and Shape of NPs

Ag, Au, and Ag/Au BNPs synthesis was determined by using different factors such as pH, AgNO_3_/HAuCl_4_·3H_2_O concentration, HH plant extract concentration, temperature, time, and stability time. To study the effect of basic and acidic conditions, pH of the reaction mixture ranged from 4 to 12 by using NaOH and HCl (0.1 M) solution. To study the effect of AgNO_3_ and HAuCl_4_·3H_2_O salt concentration, its concentration varied to 0.25, 0.5, 1, and 1.5 mM. To study the effect of HH extract concentration on NPs synthesis, its concentrations varied from 0.5, 1, 1.5, and 2 mL. To study the temperature effect, the NPs synthesis was carried out at different reaction temperature ranges (20, 40, 60, 80, and 100 °C). To study the rate of completion of the reaction, NPs were synthesized at different time intervals (1, 2, 3, and 24 h). The synthesized NPs’ stability was studied after 1 day, 3 months, and 6 months. 

### 3.5. Characterization

In the aqueous solution, Ag, Au, and Ag/Au BNPs’ concentrations were confirmed by using a SHIMADZU UV SPECTROPHOTOMETER (UV-1800). The purified Ag, Au, Ag/Au BNPs, and HH plants extract were examined for the presence of different phytochemicals by using a Fourier Transform-Infrared (FT-IR) Shimadzu (IR Prestige-21) spectrometer (Japan). The crystalline nature of the Ag, Au, and Ag/Au BNPs was determined by using the JDX-3532 (JEOL JAPAN) X-ray diffractometer (XRD) with λ-1.54 A° wavelength. The size and shape of Ag, Au, and Ag/Au BNPs were determined by using JEOL Scanning Electron Microscope (SEM) Model JSM-5910 (Japan). The presence of elements in synthesized Ag, Au, and Ag/Au BNPs was determined by using electron diffraction X-ray spectroscopy (EDX).

### 3.6. Antibacterial Activity (Agar Diffusion Method)

Antimicrobial activities of AgNO_3_ salt, HAuCl_4_·3H_2_O salt, HH extracts, HH-AgNPs, HH-AuNPs, and BNPs were determined according to the procedure [32]. Bacterial suspensions were prepared by caring 1 CFU of bacterial strain from the maintained slants in 0.9% NaCl solution, followed by incubation at 37 °C for 24 h. Nutrient agar (2.8 g/100 mL dH_2_O), autoclaved at 121 °C for 15 min. About 25 mL media (cooled to room temperature) were added to each Petri plate and then kept back for solidification. By swab, a good amount of each bacterial suspension was taken and streaked on the Petri plates. Per plate, 5 wells (3–6 mm) were dug with a sterilized cork borer, with 1 central hole surrounded by 4 holes. All of the wells were properly labeled. DMSO (20 µL, as well as AgNO_3_ salt, HAuCl_4_·3H_2_O salt, HH extracts, HH-AgNPs, HH-AuNPs, and BNPs (concentrations of 30, 60, and 90 µg/mL), were added to the surrounding holes. Levofloxacin (final concentration of 1 mg/mL in DMSO) was added to the central holes. DMSO served as a negative control. Zones of inhibition were measured after a day with a graduated ruler. A clear zone of bacterial inhibition was observed around the holes. The diameter of the clear zone was measured in mm. Inhibition by active ingredients of the extract was determined by measuring linear growth (mm) in Petri plates concerning levofloxacin (mm), a standard antibiotic (positive control).

### 3.7. Antifungal Bio-Assay

SDA media (6.5 g/100 mL dH_2_O) were autoclaved at 121 °C for 15 min and kept at room temperature for cooling (40–50 °C). To each test tube, 7 mL media, along with 67 µL AgNO_3_ salt, HAuCl_4_·3H_2_O salt, HH extracts, HH-AgNPs, HH-AuNPs, and BNPs from the concentrations 20, 40, 60, 80, and 100 µg/mL were added, and kept in slant position for solidifying inside the laminar flow hood. DMSO served as negative control and terbinafine as a positive control. Then, the particular fungal strains (*A. niger*, *A. flavus*, and *A. fumigatus*) were homogeneously spotted in each test tube. The test tubes were then sealed with cotton plugs. The entire setup was placed in an incubator with open water at 30 °C. Growth inhibition was determined by measuring linear growth [32].

### 3.8. Statistical Analysis

All the data were expressed as mean ± SE. Normal data distribution was confirmed.

## 4. Discussion

In nanoscience, the design and development of nanomaterials with unusual optoelectronic and physicochemical features act as a cornerstone. The present study discloses the bioreduction properties of Ag and Au ions into Au, Ag, and Ag/Au BNPs using ethnopharmacologically important HH plant extract. In the present investigation, Ag and Au NPs were synthesized from AgNO_3_ and HAuCl_4_·3H_2_O, while Ag/Au BNPs were manufactured by treating the HH extract with a mixture of both AgNO_3_/HAuCl_4_·3H_2_O (1:1). The synthesis of NPs was monitored visually with a change in the color of reaction mixtures under optimum conditions. After 24 h, the NPs became stable and no more synthesis was detected, which was monitored with UV–vis spectroscopy. These results are supported by the [33,34] report of bimetallic alloys.

To accomplish maximum production of NPs, different intrinsic factors, such as pH, salt concentrations, plant concentration, time, temperature, and stability period were studied. The present results justify the earlier report [35] and confirm the effect of different intrinsic factors on the maximum synthesis of NPs. An ideal pH is required for the synthesis of controlled shape/size NPs [36,37]. As a result of extensive screening with different pH, syntheses of Ag, Au, and Ag/Au BNPs were successful with pH 12, 4, and 6, respectively. These results showed that AgNPs’ synthesis is supported by basic conditions and suppressed by acidic conditions; AuNPs’ is supported by highly acidic conditions, and Ag/Au BNPs’ synthesis is supported by nearly neutral conditions [23,38]. In the current study, NPs’ reduction and capping were achieved with the different AgNO_3_, HAuCl_4_·3H_2_O salt, and HH plant concentrations. Screening with various concentrations of AgNO_3_, HAuCl_4_·3H_2_O salt, and plant synthesis of Ag, Au, and Ag/Au BNPs was successful, with 1 mM salt and 1 mL plant concentration. Medicinal plants are a rich source of phytochemicals that act as reducing, capping, and mediators for NPs. However, the composition of these active secondary metabolites varies from plant to plant, dependent on the nature, part, type of plant, and method followed for the extraction of these metabolites [39]. Thus, a slight increase in the HH extract and slat concentration beyond 1 mL and 1 mM, respectively, increases the wavelength. Thus, this slight variation in the absorbance values is because of the change in the particle size. The results of the present research are also supported by [23], who reported an absorbance at 414 nm for the 1 mL *Calligonum polygonoides* and 1 mM salt with high absorbance of 2.375. The results of the present research work are also supported by [31], who reported the synthesis of NPs by using *Gloriosa superba* leaf extract. The Ag/Au BNPs showed an absorption peak at 542 nm, which is in good agreement with the previous results of the BNPs (Ag:Au) 1:1 ratio [30]. The temperature of the reaction medium determines the nature of NPs [40]. In this study, NPs were synthesized at varying temperature ranges (20 to 100 °C). At a lower temperature of 20 °C, a broad peak with low absorbance was obtained, but with an increase in temperature up to 40 °C, a sharp peak was recorded. With a further increase in temperature, the peak become broader and broader and final broad peaks were revealed for Ag, Au, and Ag/Au BNPs at the highest temperature of 100 °C. Thus, from these changes in peaks with temperature, it is revealed that NPs synthesis required an optimal temperature for its stability [23,41,42]. The quality, size, shape, and type of NPs are greatly influenced by incubated reaction time [43]. After mixing the HH extract with 1 mM salt, no color change was observed after 1 h, 2 h, and 3 h, but after 24 h of incubation an absorption spectrum of 417 nm, 576 nm, and 542 nm were reported, which corresponds to the Ag, Au, and Ag/Au BNPs. From the time study, it is revealed that an optimum time is required for stable NPs synthesis [23,44]. The reactivity and potency of the NPs greatly depends on their stability. The NPs have an optimum stability time, after which their potency is decreased or completely lost. In the present research, the stability of the NPs was checked at different periods of 1 day, 3 months, and 6 months. Sharp peaks at 17 nm, 576 nm, and 542 nm, which correspond to the Ag, Au, and Ag/Au BNPs, appeared after incubation in the dark for 1 day to 3 months. However, after 6 months the absorption band became broad and showed low absorbance. The present research study indicated that NPs remained stable for about 3 months, but after 6 months they lost their stability. The results are best supported by [45], who used *Kiwifruit* juice at room temperature for NPs’ synthesis and revealed NPs in the range of 5 to 25 nm which remain stable for more than one month. Mean particle size and stability of NPs increased at 72 h of stirring time [44]. Thus, the variations in the reaction time may occur due to reasons such as: particle aggregation due to long time storage; particles shrinking or growing during long storage; the shelf life of particles, and so forth. All of these affect the reactivity and potential of the particles [46].

The phyto-components present in aqueous plant extract that are responsible for stabilizing and mediating the NPs were illustrated using FTIR analysis. The scientific studies reported that FTIR is ideal for forecasting functional moieties. In the present study, various vibrational stretches occurred at different peaks, which correspond to polyhydroxy, phenol, carboxyl, proteins, lipids, amide, alkynes, alkene, etc. The FTIR analysis delineated the different functional groups, such as hydroxyl, ketones, carboxyl, amino, alkaloids, amide, etc., that are responsible for the induction of metal ions to NPs [47,48]. The present result of FTIR analysis is justified by the prior findings of [49]. Stimulatingly, in plant-induced NPs, the phytochemicals of the plant played a significant role in the stabilization of NPs, which is critical for transcription and its applicative properties. These results are also supported by the earlier reports of [50,51].

The XRD peaks of Ag, Au, and Ag/Au BNPs exhibited four brag reflections which agree with the face-centric cube crystalline nature of Ag, Au, and Ag/Au BNPs. The well-resolved and strong pattern of the XRD showed that both Ag and Au NPs that are formed due to the reduction of Ag+ and Au+ ions are crystal-like. The Ag/Au BNPs that are formed by the combined reduction of Ag and Au ions are not different from the Ag and Au monometallic NPs, which can be similarly described as a crystal lattice structure. The mean calculated sizes were found to be 13.3, 10.72, and 8.34 nm for Ag, Au, and Ag/Au BNPs, respectively. The obtained results are justified by the earlier reports of [31,52].

The EDX analysis was used for the confirmation of elemental composition, purity, and relative abundance of NPs [53,54]. The percentages of Ag, Au, and Ag/Au metal in the present study were found to be appreciable, such as the Ag signal of the AgNPs with 22.75%, Au signal of the AuNPs with 48.08%, and Ag signal with 12%, and Au signal with 38.26% of the Ag/Au BNPs. The other elements which are shown by EXD analysis served as capping organic agents which are bounded on the NPs surface [55,56].

The surface morphology of NPs was studied by SEM analysis. In the present study, the SEM analysis showed an image of high density, spherical, and monodispersed NPs. The white individual spots represent the NPs, while the longer spots are the collection of NPs in the SEM photograph. The NPs from the SEM photograph have been observed with a diameter of 40, 30, and 20 nm of Ag, Au, and Ag/Au BNPs, respectively. The capping agent of the plant extract indicates the stabilization of the NPs because they were not in direct contact even in the aggregated form. The larger NPs during SEM measurements may be due to the combination of the smaller NPs. The obtained results are justified by the earlier reports of [57,58]. 

Different pathogenic bacteria cause universal infections: *S. aureus* causes skin infections, pneumonia, osteomyelitis, and endocarditis; *K. pneumonia* causes bloodstream infections, wound pneumonia, infections at surgical site, and meningitis; *A. meriye* can result in abscesses, pain, and inflammation; *S. pyogenes* causes pneumonia, necrotizing fasciitis, myonecrosis, scarlet fever, and bacteremia; *E. coli* causes cholangitis, cholecystitis, infection in urinary tract, bacteremia, and traveler’s diarrhea; *S. marcescens* causes severe infections, bacteremia, pneumonia, infection in urinary tract, infection in biliary tract, meningitis, wound infection, and endocarditis, a foodborne pathogen; *B. cereus* can produce toxins that result in two types of gastrointestinal infection: the emetic (vomiting) and the diarrhea syndrome; MRSA causes severe skin infections; *M. luteus* is an adaptable pathogen that causes endocarditis, meningitis, septic arthritis, in HIV positive patients causes chronic cutaneous infections and catheter infections; and *S. pneumoniae* causes various infections such as pneumonia, sinusitis, osteomyelitis, otitis media, septic arthritis, bacteremia, and meningitis [59,60,61]. Due to the resistance of microbes to drugs, researchers are looking at the development of innovative agents against bacteria [62]. The NPs synthesized by the green approach are applied extensively in different biomedical applications [63]. The NPs kill microorganisms due to their interaction with the bacterial cell membranes. The NPs, such as the Ag, Au, and Ag/Au BNPs, exhibited the lowest noxiousness and thermal and pH resistance, and thus proved to be an outstanding antibacterial agent that could be useful in biomedical applications [64]. The green-synthesized Ag, Au, and Ag-Au NPs and HH plant extract displayed promising antibacterial activity against *A. meriye*, *S. pyogenes*, *B. cereus*, *S. aureus*, *MRSA*, *M. luteus*, *S. pneumonia*, *K. pneumonia*, *E. coli*, and *S. marcescens* in a dose-dependent manner. In this study, the appreciable antibacterial potential of Ag/Au BNPs compared to MNPs depends on morphology, surface area, particle size and shape, and surface polarity [65].

Like bacteria, different pathogenic fungi also cause systemic illness of the mouth, skin, lungs, blood, and liver, as well as causing hypersensitive reactions, tinea cruris, and athlete’s foot [66,67,68,69]. For the therapeutic management of invasive systemic fungal infections, only about 10 antifungal drugs are permitted in the United States of America by the Food and Drug Administration (FDA) authority [66,67]. From the present observation, the highest fungicidal activity was observed with Ag/Au BNPs when compared with HH plant, Ag, and Au NPs at a similar concentration. The significant antifungal effect of Ag/Au BNPs compared to single metal Ag/Au NPs is due to their small size; as confirmed by investigators, the smaller the size of the particle, the greater the inhibition [70,71]. In comparison, [72] has analyzed the antifungal effect of NPs against species of aspergillus, revealing potential to control fungal growth. Moreover, [73] could be able to gain an antifungal effect against Aspergillus species at 80 µg/mL concentration of NPs. This outcome is supported by the aforementioned research, which was accompanied by [74], who reported that the NPs’ interaction with lysed cells’ intracellular substances caused their coagulation and the particles were thrown out of the liquid system. The Au and Ag ions mechanism of inhibition action on micro-organisms demonstrates that the microbe treatment by metal ions prompts loss of DNA, its capability replication and translation, as well as other cellular enzymes and proteins that are necessary for the production of ATP (coenzyme adenosine triphosphate), resulting in living cell inactivation of [75]. It has also been assumed that metal NPs mainly affect the function of membrane-bound enzymes in the chain of respiration [76].

## 5. Conclusions

In this study, medicinally active phytochemical-induced HH-AgNPs, HH-AuNPs, and Ag/Au BNPs were synthesized from HH extract, an indigenous plant found in abundance in Pakistan. These NPs were characterized by the following techniques: UV-vis spectroscopy, FTIR, XRD, EDX, and SEM. For antimicrobial activity, Ag and Au have always been excellent choices. Finally, this investigation revealed that Ag/Au BNPs have better antimicrobial activity than HH plant extract, HH-AgNPs, and HH-AuNPs. This work assimilates microbiology and nanotechnology, thus leading to possible advances in antimicrobial agents’ formulation. However, future studies on the antimicrobial effect of these NPs on microbes are essential to completely assess its potential use as a novel biocidal material.

## Figures and Tables

**Figure 1 molecules-27-07895-f001:**
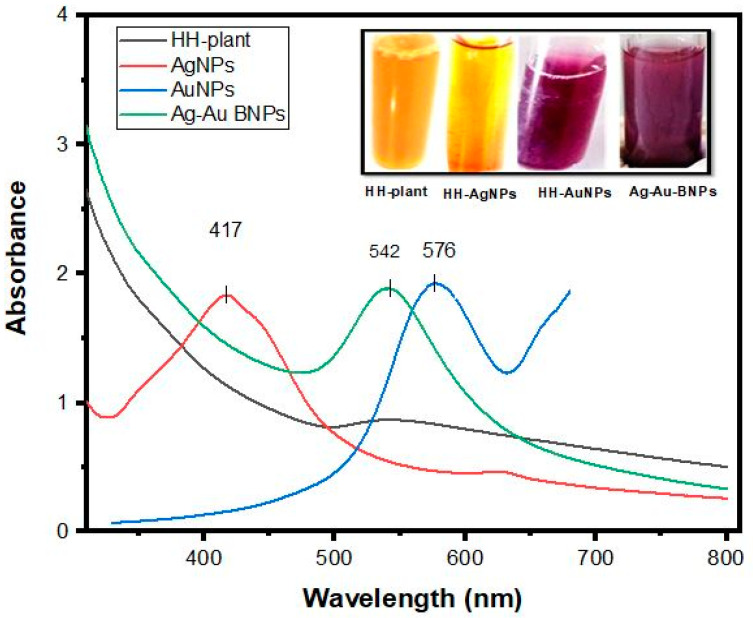
UV-vis spectrum of biologically synthesized NPs by using the HH plant extract. The absorbance peaks are 417 for HH-AgNPs, 576 for HH-AuNPs, and 542 for Ag/Au BNPs.

**Figure 2 molecules-27-07895-f002:**
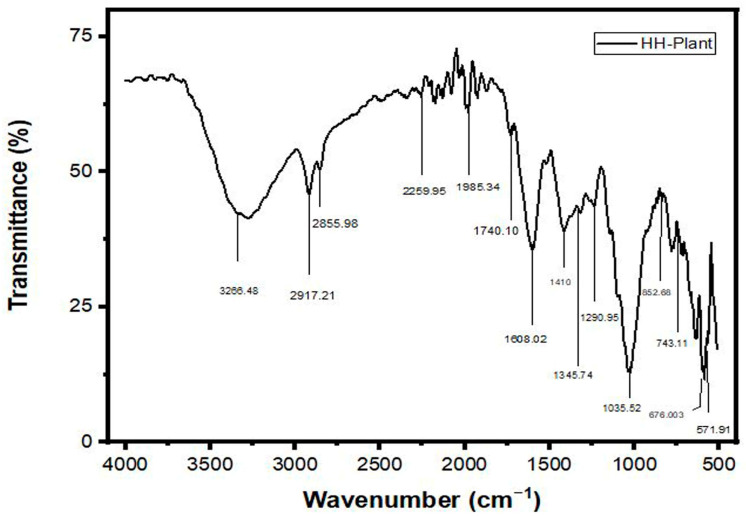
FT−IR analysis of HH plant extract.

**Figure 3 molecules-27-07895-f003:**
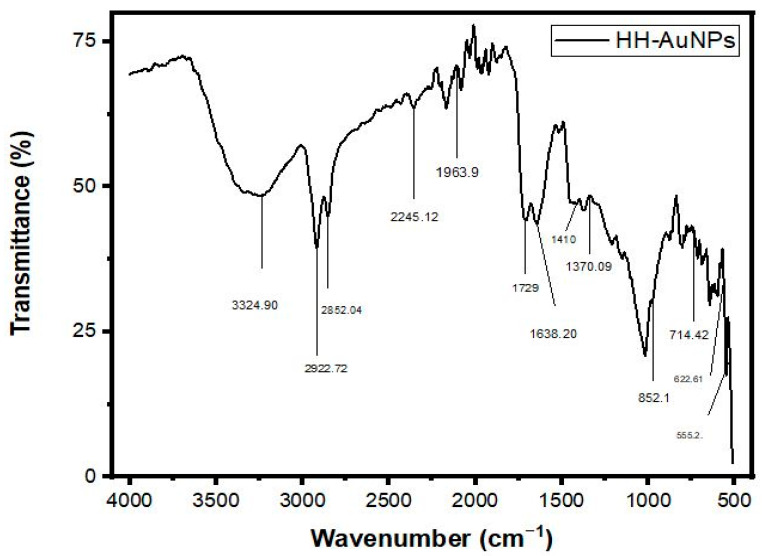
FT−IR analysis of HH-AuNPs.

**Figure 4 molecules-27-07895-f004:**
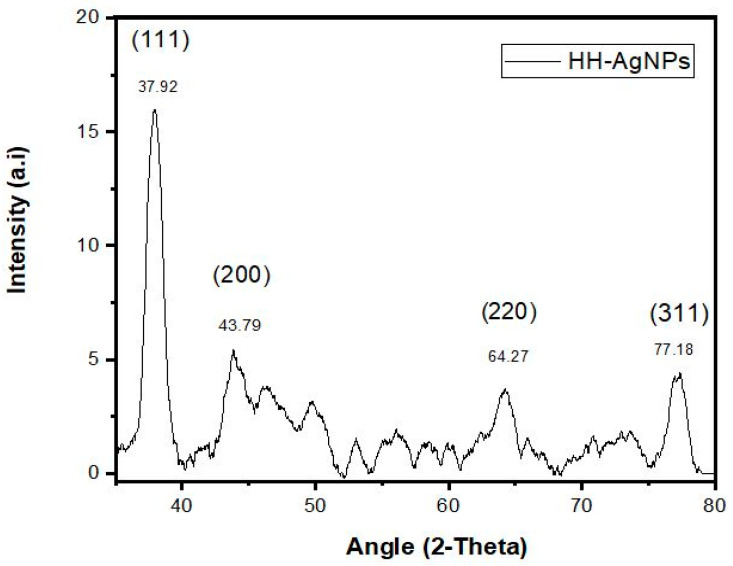
XRD pattern analysis of biologically synthesized AgNPs.

**Figure 5 molecules-27-07895-f005:**
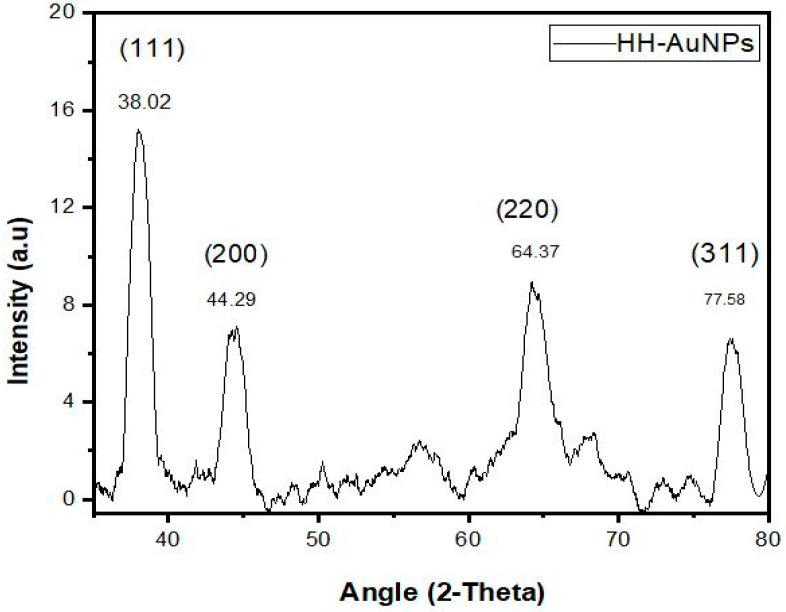
XRD pattern analysis of biologically synthesized AuNPs.

**Figure 6 molecules-27-07895-f006:**
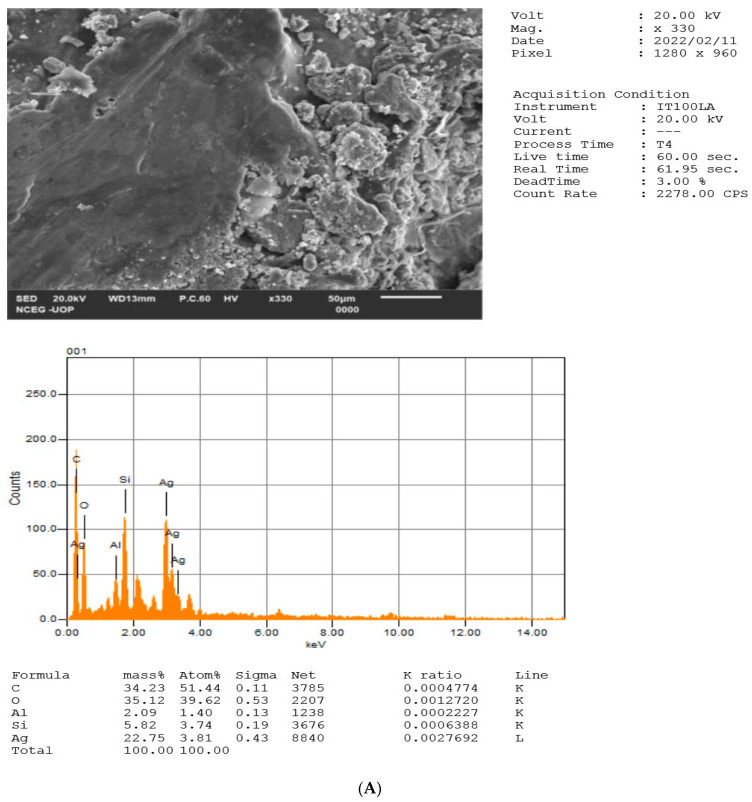
EDX pattern of biologically synthesized NPs (**A**) HH-AgNPs, (**B**) HH-AuNPs, and (**C**) Ag/Au BNPs.

**Figure 7 molecules-27-07895-f007:**
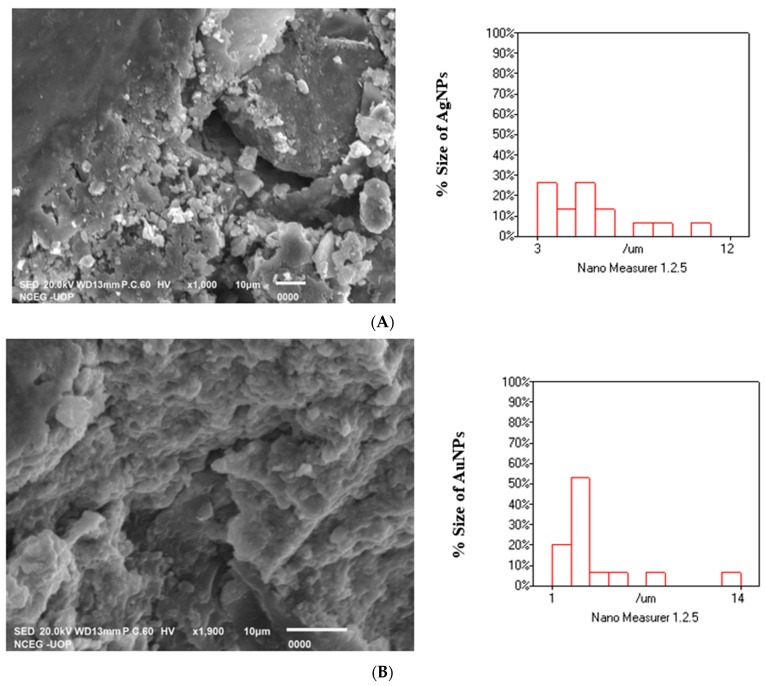
SEM images of biologically synthesized NPs (**A**) HH-AgNPs, (**B**) HH-AuNPs, and (**C**) Ag/Au BNPs.

**Table 1 molecules-27-07895-t001:** FTIR interpretation of compounds in HH whole plant extract, HH-AgNPs, HH-AuNPs, and HH-BNPs.

S. No	Wave Number cm^−1^ [Reference Article]	Wave Number cm^−1^ [HH-Plant]	Wave Number cm^−1^ [HH-AgNPs]	Wave Number cm^−1^ [HH-AuNPs]	Wave Number cm^−1^ [HH-BNPs]	Functional Group Assignment	Phyto Compounds Identified
1	3400–3200	3266.48	3352.52	3324.90	3338.71	O-H stretch	Poly hydroxy compound
2	2935–2915	2917.21	2915.41	2922.72	2915.41	Asymmetric stretching of –CH (CH_2_) vibration	Saturated aliphatic compound lipids
3	2865–2845	2855.98	2858.54	2852.04	2852	Symmetric stretching of –CH (CH_2_) vibration	Lipids, protein
4	2260–2100	2259.95	2258.93	2245.12	2252.43	Carbon-carbon triple bond	Terminal alkynes
5	2100–1800	1985.34	1977.009	1963.9	1963.19	Carbonyl compound frequency	Transition metal carbonyls
6	1740–1725	1740.10	1730.01	1729	1737.32	C=O stretch	Aldehyde compound
7	1650–1600	1608.02	1617.08	1638.20	1624.39	C=O stretching vibration, Ketone group	Ketone compound
8	1410–1310	1410	1412.34	1410	1410	O-H bend, alcoholic group	Phenol or tertiary alcohol
9	1340–1250	1290.95	1277	1250	1251	CN stretch	Aromatic primary amine
10	995–850	852.68	850.01	852.1	850	P-O-C stretch	Aromatic phosphates
11	800–700	743.11	728.2	714.42	735.66	C-Cl stretch	Aliphatic chloro compound
12	700–600	676.003	664.86	622.61	664.86	C-Br stretch	Aliphatic bromo compounds
13	690–550	571.91	580.36	555.2	573.05	Halogen compounds (bromo compounds)	Aliphatic bromo compounds

**Table 2 molecules-27-07895-t002:** Determination of different parameters of Ag, Au, and Ag/Au BNPs by XRD analysis.

**HH-AgNPs**					
S. no	2θ Value	Element	Plane	Interplanar spacing (d)	Lattice constants (a_0_)
1	37.92	Ag	1 1 1	2.36 Å	4.08 Å
2	43.79	Ag	2 0 0	2.06 Å	4.12 Å
3	64.27	Ag	2 2 0	1.44 Å	4.07 Å
4	77.18	Ag	3 1 1	1.23 Å	4.07 Å
**HH-AuNPs**					
S. no	2θ Value	Element	Plane	Interplanar spacing (d)	Lattice constants (a_0_)
1	38.02	Au	1 1 1	2.36 Å	4.08 Å
2	44.29	Au	2 0 0	2.06 Å	4.08 Å
3	64.37	Au	2 2 0	1.44 Å	4.07 Å
4	77.58	Au	3 1 1	1.22 Å	4.04 Å
**HH-BNPs**					
S. no	2θ Value	Element	Plane	Interplanar spacing (d)	Lattice constants (a_0_)
1	38.12	Ag-Au	1 1 1	2.36 Å	4.08 Å
2	44.09	Ag-Au	2 0 0	2.06 Å	4.12 Å
3	64.76	Ag-Au	2 2 0	1.44 Å	4.07 Å
4	77.48	Ag-Au	3 1 1	1.23 Å	4.07 Å

**Table 3 molecules-27-07895-t003:** Shows different elemental composition in EDX analysis.

S. no	Weight %			Atomic %		
Elements	HH-AgNPs	HH-AuNPs	HH-BNPs	HH-AgNPs	HH-AuNPs	HH-BNPs
C	34.23	46.34	23.28	51.44	90.58	56.47
O	35.12	nill	11.63	39.62	nill	21.19
Al	2.09	nill	nill	1.40	nill	nill
Si	5.82	nill	nill	3.74	nill	nill
Cl	nill	5.58	12.02	nill	3.69	9.88
Na	nill	nill	2.81	nill	nill	3.56
Ag	22.75	nill	12	3.81	nill	3.24
Au	nill	48.08	38.26	nill	5.73	5.66

**Table 4 molecules-27-07895-t004:** Results of preliminary antibacterial assay.

**A. HH plant extract zone inhibition (mm)**										
**Concentration (** **µg)**	** *S. aureus* **	** *K. pneumonia* **	** *A. meriye* **	** *S. pyogen* **	** *E. coli* **	** *S. marcescens* **	** *B. cereus* **	** *MRSA* **	** *M. luteus* **	** *S. pneumonia* **
30	0	0	0	0	0	0	0	0.5 ± 0.04	0	0
60	0	1.5 ± 0.13	1 ± 0.011	0	2 ± 0.012	0	0	5.5 ± 0.012	0	0
90	8.5 ± 0.04	2.5 ± 0.11	3.5 ± 0.012	0	4.5 ± 0.011	3.5 ± 0.012	5.5 ± 0.01	6.5 ± 0.011	7 ± 0.014	4 ± 0.011
Levofloxacin	13 ± 0.14	14.5 ± 0.006	11.5 ± 0.03	19 ± 0.011	15.5 ± 0.013	12 ± 0.011	9.5 ± 0.012	13 ± 0.011	11 ± 0.013	10.5 ± 0.008
DMSO	0	0	0	0	0	0	0	0	0	0
**B. HH-AgNPs zone inhibition (mm)**										
**Concentration (µg)**	** *S. aureus* **	** *K. pneumonia* **	** *A. meriye* **	** *S. pyogen* **	** *E. coli* **	** *S. marcescens* **	** *B. cereus* **	** *MRSA* **	** *M. luteus* **	** *S. pneumonia* **
30	0	0	2 ± 0.03	0	8.5 ± 0.11	6.5 ± 0.06	0	1 ± 0.11	2 ± 0.03	0
60	5.5 ± 0.02	2 ± 0.21	7.4 ± 0.11	8 ± 0.41	3 ± 0.17	7 ± 0.03	0	8.5 ± 0.22	3.5 ± 0.21	0
90	10.5 ± 0.03	5.5 ± 0.1	9.5 ± 0.1	6 ± 23	6.5 ± 0.17	3.5 ± 0.12	7 ± 0.03	6 ± 0.21	8.5 ± 0.23	8.5 ± 0.11
Levofloxacin	13.5 ± 0.01	11 ± 0.3	18.5 ± 0.3	32 ± 22	11 ± 0.11	11.5 ± 0.13	9.5 ± 0.11	17 ± 0.03	11 ± 0.05	11 ± 0.03
DMSO	0	0	0	0	0	0	0	0	0	0
**C. HH-AuNPs zone inhibition (mm)**										
**Concentration (µg)**	** *S. aureus* **	** *K. pneumonia* **	** *A. meriye* **	** *S. pyogen* **	** *E. coli* **	** *S. marcescens* **	** *B. cereus* **	** *MRSA* **	** *M. luteus* **	** *S. pneumonia* **
30	0.5 ± 0.22	0	2.5 ± 0.12	0	3 ± 0.11	0.5 ± 0.14	0	0.8 ± 0.13	0	0
60	9 ± 0.21	3.5 ± 0.11	5.5 ± 0.06	3.5 ± 0.12	5.4 ± 0.21	5.5 ± 0.11	0.5 ± 0.22	3.1 ± 0.4	6 ± 0.5	0
90	14 ± 0.22	9 ± 0.21	6.5 ± 0.05	9 ± 0.23	7.5 ± 0.12	7 ± 0.15	2.5 ± 0.11	7.5 ± 0.13	8 ± 0.11	5.5 ± 0.11
Levofloxacin	16 ± 0.12	15.5 ± 0.32	12 ± 0.11	18 ± 0.12	10 ± 0.22	12.5 ± 0.21	10 ± 0.14	16 ± 0.12	9.5 ± 0.13	10 ± 0.14
DMSO	0	0	0	0	0	0	0	0	0	0
**D. HH-BNPs zone inhibition (mm)**										
**Concentration (µg)**	** *S. aureus* **	** *K. pneumonia* **	** *A. meriye* **	** *S. pyogen* **	** *E. coli* **	** *S. marcescens* **	** *B. cereus* **	** *MRSA* **	** *M. luteus* **	** *S. pneumonia* **
30	1.2 ± 0.21	3.5 ± 0.13	2.5 ± 0.13	2 ± 0.11	3.6 ± 0.12	4 ± 0.13	0	1.5 ± 0.21	3 ± 0.32	0
60	10 ± 0.11	4 ± 0.11	6 ± 0.28	4.3 ± 0.07	5.9 ± 0.11	12 ± 0.23	9 ± 0.11	6 ± 0.31	6.5 ± 0.23	0.4 ± 0.11
90	15 ± 0.12	9.6 ± 0.12	11.5 ± 0.11	10 ± 0.08	7.8 ± 0.01	12.6 ± 0.11	9.5 ± 0.13	8.5 ± 0.11	10 ± 0.34	9 ± 0.11
Levofloxacin	11 ± 0.31	12.5 ± 0.21	14 ± 0.11	13 ± 0.18	13 ± 0.03	14 ± 0.11	15.5 ± 0.15	12.5 ± 0.15	14 ± 0.11	14.5 ± 0.13
DMSO	0	0	0	0	0	0	0	0	0	0
**E. AgNO_3_ salt zone inhibition (mm)**										
**Concentration (µg)**	** *S. aureus* **	** *K. pneumonia* **	** *A.meriye* **	** *S. pyogen* **	** *E. coli* **	** *S. marcescens* **	** *B. cereus* **	** *MRSA* **	** *M. luteus* **	** *S. pneumonia* **
30	0	0	0	1.5 ± 0.14	1.1 ± 0.11	0.5 ± 0.11	0	0	0	0
60	0	0	2 ± 0.11	2 ± 0.2	4 ± 0.11	1 ± 0.21	0	0	3.5 ± 0.11	0
90	0	0	3 ± 0.11	3 ± 0.11	5.5 ± 0.11	2 ± 0.13	0	0.2 ± 0.11	4 ± 0.11	0
Levofloxacin	14 ± 0.11	15 ± 0.06	11.5 ± 0.21	10 ± 0.18	14.5 ± 0.03	11 ± 0.21	10 ± 0.11	21 ± 0.12	15.5 ± 0.11	11.5 ± 0.09
DMSO	0	0	0	0	0	0	0	0	0	0
**F. HAuCl_4_·3H_2_O salt zone inhibition (mm)**										
**Concentration (µg)**	** *S. aureus* **	** *K. pneumonia* **	** *A. meriye* **	** *S. pyogen* **	** *E. coli* **	** *S. marcescens* **	** *B. cereus* **	** *MRSA* **	** *M. luteus* **	** *S. pneumonia* **
30	0	0	0	1.5 ± 0.14	0	0	0	0	0	0
60	0	0	0	0.5 ± 0.2	0	0	0	0	0	0
90	0	0	0	2 ± 0.11	0	0	0	0	0	0
Levofloxacin	11 ± 0.18	16.5 ± 0.16	9 ± 0.23	13.5 ± 0.12	12 ± 0.13	14 ± 0.13	10 ± 0.14	21.5 ± 0.11	10 ± 0.14	9.5 ± 0.19
DMSO	0	0	0	0	0	0	0	0	0	0

**Table 5 molecules-27-07895-t005:** Results of antifungal assay.

**A. HH-plant extract inhibition (mm)**								
		20 µg	40 µg	60 µg	80 µg	100 µg	Terbinafine	DMSO
1	*A. niger*	15 ± 0.13	53 ± 0.12	55 ± 0.11	58 ± 0.1	67 ± 0.13	145 ± 0.12	0
2	*A. fumigatus*	43 ± 0.13	64 ± 0.14	75 ± 0.08	64 ± 0.12	70 ± 0.11	145 ± 0.09	0
3	*A. flavus*	0	32 ± 0.05	58 ± 0.02	70 ± 0.11	80 ± 0.08	145 ± 0.14	0
**B. HH-AgNPs inhibition (mm)**								
		20 µg	40 µg	60 µg	80 µg	100 µg	Terbinafine	DMSO
1	*A. niger*	54 ± 0.11	56.5 ± 0.09	62 ± 0.21	67 ± 0.11	75 ± 0.13	145 ± 0.12	0
2	*A. fumigatus*	78 ± 0.21	91 ± 0.15	96 ± 0.23	97 ± 0.22	111 ± 0.15	145 ± 0.09	0
3	*A. flavus*	65 ± 07	74 ± 0.12	79 ± 0.11	83.5 ± 0.21	92.5 ± 0.13	145 ± 0.14	0
**C. HH-AuNPs inhibition (mm)**								
		20 µg	40 µg	60 µg	80 µg	100 µg	Terbinafine	DMSO
1	*A. niger*	57.5 ± 0.11	67.5 ± 0.13	69.5 ± 0.15	77.5 ± 0.11	82.5 ± 0.18	145 ± 0.12	0
2	*A. fumigatus*	65 ± 0.12	80 ± 0.09	85 ± 0.18	88 ± 0.13	94.5 ± 0.21	145 ± 0.09	0
3	*A. flavus*	45 ± 0.13	63 ± 0.06	70 ± 0.12	77 ± 0.09	92 ± 0.04	145 ± 0.14	0
**D. HH-BNPs inhibition (mm)**								
		20 µg	40 µg	60 µg	80 µg	100 µg	Terbinafine	DMSO
1	*A. niger*	65 ± 0.11	81 ± 0.13	86 ± 0.15	89 ± 0.11	93 ± 0.18	145 ± 0.12	0
2	*A. fumigatus*	69 ± 0.12	85 ± 0.09	92 ± 0.18	97 ± 0.13	114 ± 0.21	145 ± 0.09	0
3	*A. flavus*	67.5 ± 0.13	71.5 ± 0.06	85 ± 0.12	89 ± 0.09	112 ± 0.04	145 ± 0.14	0
**E. AgNO_3_ salt inhibition (mm)**								
		20 µg	40 µg	60 µg	80 µg	100 µg	Terbinafine	DMSO
1	*A. niger*	0	10 ± 0.13	19 ± 0.11	23 ± 0.12	15 ± 0.11	145 ± 0.12	0
2	*A. fumigatus*	7 ± 0.12	15 ± 0.22	22 ± 0.18	24 ± 0.14	31 ± 011	145 ± 0.09	0
3	*A. flavus*	6 ± 0.11	10.5 ± 0.26	21.5 ± 0.12	28 ± 0.12	33 ± 0.14	145 ± 0.14	0
**F. HAuCl_4_·3H_2_O salt inhibition (mm)**								
		20 µg	40 µg	60 µg	80 µg	100 µg	Terbinafine	DMSO
1	*A. niger*	0	0	0	5 ± 0.11	11 ± 0.21	145 ± 0.12	0
2	*A. fumigatus*	10 ± 0.21	16 ± 0.23	25 ± 0.11	31 ± 0.09	37 ± 04	145 ± 0.09	0
3	*A. flavus*	4 ± 0.11	10 ± 0.17	15 ± 0.13	21 ± 0.22	27 ± 0.13	145 ± 0.14	0

## Data Availability

All the relevant data is provided in the article.

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
