# Peer review of "Comparative Study of Antimicrobial Activity of Silver, Gold, and Silver/Gold Bimetallic Nanoparticles Synthesized by Green Approach"

_molecules, 2022, doi:10.3390/molecules27227895_

Round 1
Reviewer 1 Report (Previous Reviewer 2)
The authors have addressed most of the reviewers’ comments so the Ms can be accepted in its present form.
Author Response
Reviewer 1
Comments and Suggestions for Authors
The authors have addressed most of the reviewers’ comments so the Ms can be accepted in its present form.
Ans. Thank you Sir

Reviewer 2 Report (Previous Reviewer 1)
Author of the study “Comparative study of antimicrobial activity of silver, gold, and silver/gold bimetallic nanoparticles synthesized by green approach” revised the manuscript, however, the following concerns are still not resolved.
Fig. 1. Why the UV plot range is different in between samples? Please explain
Fig. 4 and 5, the baseline is incorrect. Please revise the figures and associated text thoroughly.
Fig. S7 and S8 looks similar, is it because of contamination? Please explain.
Fig. S9 is not the raw data scan of XRD. Please replace with raw data scan.
Author Response
Reviewer 2
Comments and Suggestions for Authors
Author of the study “Comparative study of antimicrobial activity of silver, gold, and silver/gold bimetallic nanoparticles synthesized by green approach” revised the manuscript, however, the following concerns are still not resolved.
Fig. 1. Why the UV plot range is different in between samples? Please explain.
Ans: . The difference in the peaks of plant and NPs is due to the reduction of the phytochemicals with the salt Ag and Au ions. (Page 4)
Fig. 4 and 5, the baseline is incorrect. Please revise the figures and associated text thoroughly.
Ans: The baseline of Fig. 4 and 5, is incorrect accordingly. (Page 8 and 9)
Fig. S7 and S8 looks similar, is it because of contamination? Please explain.
Ans: The Fig. S7 and S8 looks similar, may be due the du use of same plant extract.
Fig. S9 is not the raw data scan of XRD. Please replace with raw data scan.
Ans: Raw data scan of XRD is provided accordingly in the supplementry file.

Round 2
Reviewer 2 Report (Previous Reviewer 1)
Based on recent changes, the manuscript can be accepted for publication.
This manuscript is a resubmission of an earlier submission. The following is a list of the peer review reports and author responses from that submission.
Round 1
Reviewer 1 Report
The authors report a comparative study of the antimicrobial activity of silver, gold, and silver/gold bimetallic nanoparticles synthesized by green
approach, the work can be considered for publication after the authors should address the following comments during revision,
Please add an unprocessed XDR diffractogram in the supplementary information.
XRD pattern of the nanoparticles is too similar to each other with exact same crystalline planes in all reported nanoparticles. Although during the synthesis the same extract was used, the pattern should vary, and the analysis should be repeated. Also, for crystallite size estimation, it is suggested to first perform baseline correction.
No phosphorus was detected in EDX analysis, but it is present in FTIR analysis. The EDX analysis should be repeated to justify FTIR findings.
Line No. 65-68. Please elaborate further on the innovative (objective) part of the current research.
The green synthesis mechanism was recently explored in recent articles (doi.org/10.1016/j.jclepro.2022.133853, doi.org/10.1016/j.jhazmat.2021.126958, doi.org/10.1371/journal.pone.0274753, ) which should be discussed in the introduction to help the readers.
Figures 1 and 2C. Please provide the complete scan of all the UV data for all the samples.
Figure 8. Please provide the unprocessed FTIR scan of samples HH-BNPs and HH-AgNPs in the supplementary information.
Reviewer 2 Report
The manuscript reports on the use of herbal extracts for the Silver and Gold nanoparticles preparation and testing as antimicrobial agents.
The work is interesting, but the way of the presentation complies more to technical report. More specifically detailed Tables even for the precursors are included, a large number of Figures (UV-VIS, SEM) are listed together with a detailed description of the results in the text.
Additionally, very general expressions are used not directly related to the current work. The term “photo” is used instead of “phyto” in some cases.
Overall, the manuscript should be extensively revises with a large number of Figures and Tables (at least partially) moved to a supporting information part.